# 3’-UTR Polymorphisms of Vitamin B-Related Genes Are Associated with Osteoporosis and Osteoporotic Vertebral Compression Fractures (OVCFs) in Postmenopausal Women

**DOI:** 10.3390/genes11060612

**Published:** 2020-06-02

**Authors:** Tae-Keun Ahn, Jung Oh Kim, Hui Jeong An, Han Sung Park, Un Yong Choi, Seil Sohn, Kyoung-Tae Kim, Nam Keun Kim, In-Bo Han

**Affiliations:** 1Department of Orthopedics, CHA Bundang Medical Center, CHA University, Seongnam 13496, Korea; ajh329@gmail.com; 2Theragen Bio Co., Ltd. 145, Gwanggyo-ro, Yeongtong-gu, Suwon 16229, Korea; jokim8505@gmail.com; 3Department of Biomedical Science, College of Life Science, CHA University, Seongnam 13488, Korea; tody2209@naver.com (H.J.A.); hahnsung@naver.com (H.S.P.); 4Department of Neurosurgery, CHA Bundang Medical Center, CHA University, Seongnam 13496, Korea; nschoiuy@gmail.com (U.Y.C.); sisohn@cha.ac.kr (S.S.); 5Department of Neurosurgery, School of Medicine, Kyungpook National University, Daegu 41944, Korea; nskimkt7@gmail.com; 6Department of Neurosurgery, Kyungpook National University Hospital, Daegu 41944, Korea

**Keywords:** cobalamin, folate, homocysteine, polymorphism, osteoporosis, compression fracture

## Abstract

As life expectancy increases, the prevalence of osteoporosis is increasing. In addition to vitamin D which is well established to have an association with osteoporosis, B vitamins, such as thiamine, folate (vitamin B9), and cobalamin (vitamin B12), could affect bone metabolism, bone quality, and fracture risk in humans by influencing homocysteine/folate metabolism. Despite the crucial role of B vitamins in bone metabolism, there are few studies regarding associations between B vitamin-related genes and osteoporosis. In this study, we investigated the genetic association of four single nucleotide polymorphisms (SNPs) within the 3’-untranslated regions of vitamin B-related genes, including *TCN2* (encodes transcobalamin II), *CD320* (encodes transcobalamin II receptor), *SLC19A1* (encodes reduced folate carrier protein 1), and *SLC19A2* (encodes thiamine carrier 1), with osteoporosis and osteoporotic vertebral compression fracture (OVCF). We recruited 301 postmenopausal women and performed genotyping of *CD320 rs9426*
*C>T,*
*TCN2 rs10418*
*C>T*, *SLC19A1 rs1051296*
*G>T*, and *SLC19A2 rs16862199*
*C>T* using a polymerization chain reaction-restriction fragment length polymorphism assay. There was a significantly higher incidence of both osteoporosis (AOR 5.019; 95% CI, 1.533–16.430, *p* < 0.05) and OVCF (AOR, 5.760; 95% CI, 1.480–22.417, *p* < 0.05) in individuals with genotype *CD320* CT+TT and high homocysteine concentrations. Allele combination analysis revealed that two combinations, namely *CD320* C-*TCN2* T-*SLC19A1* T-*SLC19A2* C (OR, 3.244; 95% CI, 1.478–7.120, *p* < 0.05) and *CD320* T-*TCN2* C-*SLC19A1* G-*SLC19A2* C (OR, 2.287; 95% CI, 1.094–4.782, *p* < 0.05), were significantly more frequent among the osteoporosis group. Our findings suggest that SNPs within the *CD320* gene in 3´-UTR may contribute to osteoporosis and OVCF occurrences in some individuals. Furthermore, specific allele combinations of *CD320*, *TCN2*, *SLC19A1,* and *SLC19A2* may contribute to increased susceptibility to osteoporosis and OVCF.

## 1. Introduction

As the number of older people at risk for compromised bone health rapidly increases, identifying new risk factors for osteoporosis has become the center of attention. Researchers have reported many risk factors for osteoporosis and osteoporotic bone fractures, such as age, history of fragility fractures, smoking habits, alcohol intake, steroid use, low bone mineral density (BMD) and hyperhomocysteinemia [1]. Accumulating evidence suggests that high homocysteine level may cause osteoporosis by over production of free radicals and oxidative stress, promotion of osteoclast activity and bone resorption, and inhibition of bone formation. Vitamins are also closely related to bone health. Particularly, vitamins A, D, E and K have been proved to contribute to maintaining bone health [2]. In addition, the role of B-vitamins in bone health has been receiving increasing attention due to both individual action as vitamins and their action on influencing homocysteine concentration [3,4,5,6,7]. Deficiencies in vitamin B6, B9 (folate), and B12 (cobalamin) have been known to cause increased serum levels of homocysteine because these vitamins act as co-factors for various enzymes involved in homocysteine metabolism [1,2]. Cobalamin has been known to increase osteoblast proliferation and cobalamin deficiency may increase osteoclast formation by elevation of homocysteine and mehtylmalnonic acid levels. Folate was also reported to aid maintenance of bone density by helping to preserve optimal nitric oxide synthase activity in the bone cells [7]. Although the effects of vitamin B1 (thiamine) on bone health remains unclear, previous studies show that thiamine may inhibit receptor activator of nuclear factor κB ligand induced osteoclastogenesis, suggesting a potential link between thiamine deficiency and poor bone health [2]. 

In addition to these clinical factors, genetic variations within individuals have also been demonstrated to affect the occurrence of osteoporosis and related fractures. Many genetic association studies have confirmed the relationship between reduced BMD, candidate gene polymorphisms, and an increased risk of fracture. Some of the associated polymorphisms reside in vitamin-related genes; among these, polymorphisms within vitamin D-related genes, especially the gene encoding the vitamin D receptor, have been most widely investigated and are known to be associated with BMD in various populations [8,9]. Despite the role of B vitamins in bone metabolism, there are few studies examining the relationship between polymorphisms of vitamin B-related genes and osteoporosis. Only several homocysteine/folate metabolism-related genes have been linked with osteoporosis to date. For example, a single nucleotide polymorphism (SNPs) in methylenetetrahydrofolate reductase (*MTHFR* 677 C>T) has been shown to be associated with BMD [10,11]. Recently, it was also reported that SNPs in the 3´ untranslated regions (UTRs) of *MTHFR* (2572 C>A) and thymidylate synthase (1100 C>T) are associated with the prevalence of osteoporosis and osteoporotic vertebral compression fractures (OVCFs) [12]. 

Apart from vitamin D, B vitamins are also associated with bone metabolism. Data have suggested that B vitamins, such as folate (vitamin B9) and cobalamin (vitamin B12), affect bone metabolism, bone quality, and fracture risk in humans by contributing to homocysteine/folate metabolism [3,4,5,6]. High level of homocysteine may impair collagen cross-link within bone, thereby resulting in decreased bone mineral density and increased susceptibility to fracture [13,14]. The folate and cobalamin are important cofactors and should be transported readily to cells. *CD320* and *TCN2* genes are associated with cobalamin transport. *SLC19A1* encodes protein that transport folate. Polymorphisms of those genes may affect the homocysteine metabolism. Despite the role of B vitamins in bone metabolism, there are few studies examining the relationship between vitamin B-related genes and osteoporosis. Only several homocysteine/folate metabolism-related genes have been linked with osteoporosis to date. For example, a single nucleotide polymorphism (SNP) in methylenetetrahydrofolate reductase (*MTHFR* 677C>T) has been shown to be associated with BMD [10,11]. Recently, it was also reported that SNPs in the 3’ untranslated regions (UTRs) of *MTHFR* (2572C>A) and thymidylate synthase (1100C>T) are associated with the prevalence of osteoporosis and osteoporotic vertebral compression fracture (OVCF) [12]. 

B-vitamins should be transported readily to cells to maintain intracellular concentrations. Thiamine can be transported into mammalian cells by thiamine transporter 1, also known as thiamine carrier 1 (TC1) or soluble carrier family 19 member 2 (SLC19A2), which is encoded by the *SLC19A2* gene. Cobalamin is absorbed in the distal ileum by binding to gastric intrinsic factor. The absorbed cobalamin then binds to transcobalamin II (TC II, encoded by *TCN2* gene) within the enterocyte, and the cobalamin-TC II complex is released into the blood stream. The complex is transported to all tissues where it can be internalized into cells by binding to the TC II receptor (encoded by CD320) [15]. Transport of folate into mammalian cells can occur via folate receptor 1 (RFC1) which in humans is encoded by the *SLC19A1* gene. Therefore, we chose four well-known SNPs of B vitamins-related genes, including *TCN2* (encodes TC II), *CD320* (encodes TC II receptor), *SLC19A1* (encodes reduced folate carrier gene (RFC1)), and *SLC19A2* (encodes thiamine carrier 1) because polymorphisms of B vitamins-related genes could reduce the availability of B vitamins contributing to the risk of osteoporosis and OVCFs [16,17]. Studies on the relationship between B vitamins and gene polymorphisms are currently insufficient. To the best of our knowledge, there have been no published studies on the association between polymorphisms in vitamin B-related microRNA (miRNA) binding sites (3´-UTR) and osteoporosis and OVCFs. Therefore, in the current study, a database search with MicroSNiPer was used to identify four SNPs in miRNA binding sites within the 3´-UTRs of vitamin B-related genes: *CD320 rs9426 C>T, TCN2 rs10418 C>T, SLC19A1 rs1051296 G>T*, and *SLC19A2 rs16862199 C>T*. The minor allele frequency for each of the four SNPs was >5% in the Asian population. We then investigated the associations of these four SNPs with osteoporosis and OVCFs in Korean postmenopausal women. 

## 2. Materials and Methods 

### 2.1. Study Design

We performed a case-control study to examine the relationship between four SNPs in vitamin B-related genes, namely *CD320 rs9426*
*C>T*, *TCN2 rs10418* C>T, *SLC19A1 rs1051296* G>T, and *SLC19A2 rs16862199* C>T, and osteoporosis and OVCF risk. The research was conducted in accordance with the principles described in the Declaration of Helsinki. The institutional Review Board of CHA Bundang Medical Center approved this study (IRB number: BD2015-043), and all participants provided written informed consent.

### 2.2. Study Population 

The study group comprised 301 postmenopausal Korean women from the South Korean province of Gyeonggi-do. Postmenopausal women were recruited from the neurosurgery and orthopedic surgery departments at the CHA Bundang Medical Center. Korean postmenopausal women ≥50 years of age were eligible for inclusion in the study. The diagnosis of osteoporosis was based on Dual-energy X-ray absorptiometry (DEXA, Norland Medical Systems, White Plains, NY, USA) of the lumbar spine and hips. Diagnosis of osteoporosis was based on a BMD threshold of 2.5 standard deviations below that of a young adult (T-score, <−2.5 or lower) according to the standard World Health Organization criteria. An OVCF was diagnosed when a progressive or newly generated compression fracture was identified after low-energy trauma. An OVCF was defined as a height reduction of the vertebrae >15% in any anterior, central, or posterior portion of the vertebrae in plain radiographs [18]. The presence of an OVCF was first determined by a trained neurosurgeon or an orthopedic surgeon and was then confirmed by a radiologist. Computed tomography (CT) or magnetic resonance imaging (MRI) was conducted for subjects with suspicious findings in the plain radiographs or bone scans.

All subjects with osteoporosis met the following criteria: 1) absence of metabolic diseases, such as diffuse idiopathic skeletal hyperostosis, pituitary gland disorders, hyperthyroidism, rheumatoid arthritis, or hyperparathyroidism; 2) no use of drugs that affect bone metabolism or blood clotting, including oral anticoagulants, oral contraceptives, hormone replacement therapy, corticosteroids, calcium, vitamin D, or vitamin B; 3) absence of seronegative spondyloarthropathy; 4) no history of stroke or ischemic heart disease; 5) no prior cancer diagnosis; and 6) Korean descent. All subjects were examined using conventional X-radiography, DEXA, and MRI to evaluate the configuration and acuity of the fracture. Whole-body bone scanning was used in cases where MRI was contraindicated. 

Control subjects were recruited from individuals who visited the CHA Bundang Medical Center for routine health examinations. Subjects with lumbar spine and hip BMD T-scores >−1.0 and no spine or hip fractures were enrolled. The exclusion criteria were identical to those used for the osteoporosis group. Demographic features and comorbidities, such as hypertension, diabetes mellitus, and other cerebro- and cardiovascular diseases, were investigated in all osteoporosis and control subjects. Hypertension was defined as systolic blood pressure (SBP) >140 mmHg or diastolic blood pressure (DBP) >90 mmHg, including subjects undergoing antihypertensive treatment. Diabetes mellitus (DM) was defined as a fasting plasma glucose level >126 mg/dL, including subjects who were already diagnosed as diabetic. 

### 2.3. Blood Sample

Plasma levels of glucose (reference range, <100 mg/dL), homocysteine (reference range <12 µmol/L), folate (reference range, 3.45–13.77 ng/mL), high-density lipoprotein (HDL; reference range, >40 mg/dL), low-density lipoprotein (LDL; reference range, <130 mg/dL), triglycerides (TG; reference range, <200 mg/dL), and vitamin B12 (reference range, 211–911 pg/mL) were measured in fasting blood samples from each subject. Homocysteine was measured by fluorescent polarization immunoassay using an Abbott IMx system (Abbott Laboratories, Abbott Park, IL, USA), folate was measured by competitive immunoassay using ACS:180 (Bayer, Tarrytown, NY, USA), and vitamin B12 was measured using the Bio-Rad Quantaphase II radioassay (Hercules, CA, USA). All assays were conducted according to manufacturer’s instructions. HDL, LDL, and TG concentrations were assessed using standard hospital protocols.

### 2.4. Genetic Analyses

We collected 5 mL of peripheral blood by tube coated with anticoagulant, separated the buffy coat with leukocytes through centrifugation, and extracted genomic DNA using the G-DEXIIb kit (iNtRon Inc., Seongnam, South Korea). The experiment procedure was as follows. We mixed thoroughly 1.5 mL peripheral blood and RBC lysis solution and incubated for 5 min at room temperature. During incubation we turned it over again at least once. After 5 min, centrifugation was performed at 10,000× g for 1 min. Subsequently, the supernatant was removed except for the white blood cell pellet. After vortexing to resuspend the cell pellet in the tube, cell lysis solution was added to the resuspended cells and pipetted up and down to lyse the cells at room temperature. When the cells were sufficiently lysed, 100 μL of PPT buffer was added to the cell lysate, vortexed vigorously at high speed for 20 sec, and centrifugation was performed at 13,000× *g* for 5 min. The supernatant containing DNA was transferred to a new 1.5 mL tube, 300 μL 100% isopropanol (2-propanol) was added, and the sample was mixed by gently inverting several times. Thereafter, the supernatant was removed from the clean absorbent paper by centrifugation at 13,000× g for 1 min. In the next step, 1 mL 70% ethanol was added, the tube was inverted several times to wash the DNA pellet, and centrifuged for 1 min at 13,000× g. We removed the supernatant, inverted, drained the tube from a clean absorbent paper and air dried for 10 min. After drying sufficiently, 150 μL of DNA rehydration buffer was added. After rehydration of the DNA by incubation at 65 °C for 60 min, DNA purity was confirmed by measurement of the O.D 260: 280 ratio, and stored at −20 °C for long-term storage. We genotyped the 3’-UTR polymorphisms of *CD320 rs9426*
*C>T*, *TCN2 rs10418* C>T, *SLC19A1 rs1051296* G>T, and *SLC19A2 rs16862199* C>T SNPs by polymerase chain reaction-restriction fragment length polymorphism (PCR-RFLP) (Figure 1). The PCRs were performed using the following primers, which were designed by PrimerQuest (Integrated DNA Technologies, Coralville, IA, USA): *CD320 rs9426* C>T, forward primer 5’-TGT CTT AAG CAC AGG GCC GTT CTA-3’ and reverse primer 5’-GGT CCC TGG ACA CTC CCC ATG-3’; *TCN2 rs10418* C>T, forward primer 5’-ACT CTG TTA GAG TGG CAG ATC-3’ and reverse primer 5’-GCT TTA ATT TTG TCA GAG GCA GG-3’; *SLC19A1 rs1051296* G>T, forward primer 5’-GCT TCT CTG TCT CTG TGG AAA-3’ and reverse primer 5’-AAG CCT GGC ACA TAC CAA-3’; and *SLC19A2 rs16862199* C>T, forward primer 5’-GCA GGA ATC ACA TCT ATC CTA GTT CC-3’ and reverse primer 5’-GCT TAA GGT ACG CTT GCT TGT C-3’. The underlined bases indicate mismatches with the complementary sequence. For RFLP analysis of the SNPs, the PCR products for *CD320*, *TCN2*, *SLC19A1*, and *SLC19A2* were digested with the restriction enzymes *Sty*I, *Bgl*II, *Ban*I, and *Hpy*188I, respectively. To confirm the four SNPs and validate the RFLP results, 10%–20% of the samples were randomly selected, used for a second round of PCR, and analyzed by DNA sequencing using an automatic ABI3730xL DNA analyzer (Applied Biosystems, Forster City, CA, USA). The concordance of the quality control sample was 100%.

### 2.5. Statistical Analyses

Differences between groups were assessed using the chi-squared test and Student’s t-test for categorical variables and continuous variables, respectively. Multivariate logistic regression and Fisher’s exact test were used to compare the genotype and allele combination frequencies, respectively, between cases (osteoporosis group, including non-OVCF and OVCF subjects) and controls. Allele frequencies were evaluated for deviation from Hardy-Weinberg equilibrium using a threshold of *p* = 0.05. To estimate the relative risk for osteoporosis with respect to subject genotype in the non-OVCF and OVCF groups, the odds ratio (OR) and 95% confidence interval (CI) were calculated. Analysis of variance (ANOVA) was used to analyze the association of the genotypes with BMD, body mass index (BMI), DBP, SBP, and levels of glucose, homocysteine, folate, HDL, LDL, TG, and vitamin B12. A p-value ≤ 0.05 was considered to indicate statistical significance. ORs were adjusted (AOR) for possible confounders, such as age, sex, DM, hypertension, and serum folate and vitamin B12 levels. Statistical analyses were performed using GraphPad Prism 4.0 software (GraphPad Software, Inc., San Diego, CA, USA), MedCalc v.18.11.3 software (MedCalc Software, Mariakerke, Belgium), and HAPSTAT 3.0 (University of North Carolina, Chapel Hill, NC). We constructed all possible allele combinations for the four SNPs and analyzed gene-gene interactions using the multifactor dimensionality reduction (MDR) method (MDR software package v.2.0, www.epistasis.org) [19].

## 3. Results

### 3.1. Patient Characteristics

The mean ages in the control group (*n* = 158) and osteoporosis group (*n* = 143) were 69.36 ± 6.26 years and 69.38 ± 7.25 years, respectively. Of the 143 subjects with osteoporosis (osteoporosis group), 74 patients were diagnosed with OVCF (OCVF group). When comparing the control group with the osteoporosis group, we found that patients with osteoporosis were significantly more likely to have higher blood glucose levels, lower BMI, and decreased LDL and vitamin B12 levels. Patients in the OVCF group had significantly higher blood glucose levels, lower folate levels, and lower BMIs than subjects in the control group [9] (Table 1).

### 3.2. Genotype Frequencies of CD320, TCN2, SLC19A1, and SLC19A2 SNPs

The genotype and allele frequencies of the *CD320 rs9426*
*C>T*, *TCN2 rs10418* C>T, *SLC19A1 rs1051296* G>T, and *SLC19A2 rs16862199* C>T SNPs were compared between control subjects and osteoporosis patients with or without OVCF. There were no statistically significant differences in genotype or allele frequencies between the control and osteoporosis groups for any of the four SNPs (Figure 2; Table 2).

### 3.3. Allele Combination Analysis Using the MDR Method

We then used the MDR method to compare allele combination frequencies for the four SNPs between the control and osteoporosis groups (Table 3). The allele combinations listed below are presented according to the following gene order: *CD320, TCN2, SLC19A1,* and *SLC19A2*. The allele combination analysis revealed significant differences between the groups. Specifically, two allele combinations differed significantly between the control and osteoporosis (non-OVCF and OVCF) groups: C-T-T-C (OR 3.244; 95% CI 1.478–7.120, *p* < 0.05) and T-C-G-C (OR 2.287; 95% CI 1.094–4.782, *p* < 0.05). Five allele combinations demonstrated significant differences between the control and OVCF groups: C-C-G-T (OR 1.280; 95% CI 1.280–7.386, *p* < 0.05), C-C-T-C (OR 1.028; 95% CI 1.028–2.788, *p* < 0.05), C-C-T-T (OR 1.522; CI 1.522–15.900, *p* < 0.05), C-T-T-C (OR 1.482; 95% CI 1.482–9.187, *p* < 0.05), and T-C-G-C (OR 1.556; 95% CI 1.556–8.089, *p* < 0.05). Three allele combinations showed significant differences between the control and non-OVCF groups: C-T-G-C (OR 0.075; CI 0.075–0.887, *p* < 0.05), C-T-G-T (OR 1.098; CI 1.098–393.000, *p* < 0.05), and C-T-T-C (OR 1.133; 95% CI 1.133–6.844, *p* < 0.05). Among these combinations, combination C-T-G-T in the osteoporosis group demonstrated the highest OR (OR, 14.850; 95% CI, 0.810–272.100, *p* < 0.05). 

### 3.4. Stratified and Interactions Analyses between CD320, TCN2, SLC19A1, and SLC19A2 SNPs and Clinical Parameters 

To determine whether the four SNPs were associated with osteoporosis and OVCF prevalence in specific subsets of patients, we conducted a stratified analysis of the data according to age, hypertension, DM, and serum levels of folate, vitamin B12, and homocysteine. To enable interaction analyses for serum levels, we established cut-off values for folate, vitamin B12, and homocysteine, using cut-offs at the bottom 15% for folate (4.59 nmol/L) and B12 (395 pg/mL), and a cut-off at the top 15% for homocysteine (12.68 µmol/L).

The incidence of osteoporosis was significantly higher for genotype *CD320 rs9426* CT+TT than genotype CC both in patients ≥69 years of age (AOR 2.399, CI 1.101–5.226) and in patients with high homocysteine concentrations (homocysteine > 12.68 µmol/L) (AOR 5.019, CI 1.533–16.430, *p* < 0.05; Figure 3A). The incidence of osteoporosis was also significantly higher for genotype *TCN2* CT+TT than genotype CC in subsets of patients without hypertension, with diabetes, with low folate levels, or with low homocysteine concentrations (Table 4). To evaluate the effect of the dominant genotype in the stratified conditions of each parameter, each SNP was also analyzed using a dominant model. There were no significant differences between dominant and recessive SNPs (Appendix A). 

Next, the incidence of OVCF was analyzed with respect to each SNP. The results showed that OVCF incidence was increased for genotype *CD320 rs9426* CT+TT compared to genotype CC both in patients with low folate levels (AOR 7.307, CI 1.975–27.033) and in patients with high homocysteine concentrations (homocysteine > 12.68 µmol/L) (AOR 5.760, CI 1.480–22.417, *p* < 0.05; Figure 3B). The genotype *SLC19A1 rs1051296* GT+TT was also associated with a higher incidence of OVCF than genotype GG in patients with low folate levels (AOR 3.589, CI 1.440–8.950; Table 5). In the stratified conditions of each parameter, the dominant model did not demonstrate significant differences, although genotype *CD320 rs9426* CT+TT exhibited an increased frequency of OVCF in patients with low folate levels (AOR 2.040, CI 0.500–8.322) or high homocysteine concentrations (AOR 3.500, CI 0.795–15.400; Appendix A).

## 4. Discussion

Emerging evidence suggests that B–vitamins, in particular vitamin B1, B12 (cobalamin) and B9 (folate) exert bone–protective effects, whereas homocysteine has a detrimental effect on bone health [2,3,4,5,6,7]. It is also known that folate, vitamin B12, vitamin B6, and vitamin B2 (riboflavin) are involved in the homocysteine metabolism [20]. Therefore, folate and vitamin B12 deficiencies can cause increased serum levels of homocysteine contributing to osteoporosis and osteoporotic bone fractures. Cobalamin is first bound to cobalamin transport protein (TC II, encoded by *TCN2* gene) and TC II–cobalamin complex then enters the cells via its interaction with TC II receptor (encoded by *CD320* gene) on the cell surface [15,19,21]. Transport of thiamine and folate into cells can occur via thiamine carrier 1 (TC1) (encoded by *SLC19A2*) and reduced folate carrier (encoded by *SLC19A1*), respectively [18]. B vitamins status can be associated mainly with SNPs in genes directly involved in vitamin absorption/uptake (*CD320*) or transport (*TCN2, SLC19A1, SLC19A2*) and SNPs in these genes may have a detrimental effect on bone health by individual action as vitamins and their action on influencing homocysteine concentration [2,3,4,5,6,7]. 

In the present study, we explored possible associations between SNPs located in the 3´–UTR of the *CD320*, *TCN2, SLC19A1*, and *SLC19A2* genes and osteoporosis and OVCFs in 301 postmenopausal women. The main findings from this study are as follows:(1) Individuals with *CD320* CT+TT genotype and high homocysteine concentrations had a significantly increased risk of osteoporosis and OVCF; (2) the *CD320* C –*TCN2* T—*SLC19A1* T—*SLC19A2* C and *CD320* T–*TCN2* C–*SLC19A1* G–*SLC19A2* C allele combinations was significantly associated with an increased risk of osteoporosis. Our patients with osteoporosis, particularly those patients with OVCFs, exhibit higher blood glucose levels, decreased folate levels, and lower BMIs compared to control subjects. These findings are in accordance with results from previous osteoporosis studies. The risk for osteoporosis and osteoporotic fractures is significantly associated with both type 1 and type 2 diabetes [22,23,24]. It has also been reported that low folate levels and high homocysteine levels in serum are significantly associated with osteoporotic fractures [25]. Low BMI is an additional important risk factor for low bone mass and increased risk of osteoporotic fractures [26,27]. However, homocysteine levels did not exhibit a significant association with osteoporosis or OVCF in the current study.

A stratified analysis of osteoporosis and OVCF incidence revealed a significantly increased risk for both osteoporosis (AOR = 5.019) and OVCF (AOR = 5.760) in patients with high homocysteine levels (≥12.68 µmol/L) and genotype *CD320* CT+TT versus individuals with normal homocysteine levels and genotype *CD320* CC. These results highlight the relationship of genotypes with the incidence of osteoporosis and OVCF in individuals with high homocysteine levels; notably, there was no significant difference in homocysteine concentrations between the control and osteoporosis groups within this study (Table 1). Although a dominant model failed to identify statistical differences in the stratified analysis, a recessive model did identify an increased risk for osteoporosis and OVCF in specific subsets of individuals with the recessive genotype (Table 4 and Table 5). 

Overall, the results of the present study indicate that the incidence of osteoporosis and OVCF is significantly increased in a subset of patients who are carrying genotype *CD320* CT+TT and have high homocysteine levels. Stone et al. described that the SNP at the binding site of miR–136 is significantly associated with total homocysteine level and methylmalonic acid level [28]. The serum homocysteine concentration in normal individuals ranges from 5 to 12 µmol/L. Mild hyperhomocysteinemia is defined as a homocysteine concentration between 12–16 µmol/L [29]. In the present study, the cut–off set at the top 15%, which equated to 12.68 µmol/L, was in accordance with this established range for elevated homocysteine levels. A nationally representative, cross–sectional survey revealed a relationship between elevated homocysteine concentrations and both bone turnover markers and total body and lumbar spine BMD in women ≥50 years of age [30]. Additionally, cross–sectional data from the BPROOF study and from two Rotterdam Study cohorts have reported a significant inverse association between elevated homocysteine levels in women and both bone ultrasound parameters indicative of lower bone quality and with lower BMD. It has been hypothesized that homocysteine might impede collagen cross–link formation within bone, thereby weakening bone strength in a manner independent of BMD [31]. Yang et al. reported that mildly elevated serum homocysteine levels (>13 nmol/mL) in the general population are associated with increased risk of vertebral and hip fractures that is independent of conventional risk factors [32]. Methylation of DNA plays an important role in the regulation of gene expression, and the conversion of methionine to homocysteine involves the removal of a methyl group that could be subsequently donated to DNA. High homocysteine levels could dampen this conversion, thereby reducing DNA methylation activity and altering gene expression, which could influence expression of the *CD320* gene. 

Allele–allelic combination analysis revealed that the various combinations were significantly associated with osteoporosis and OVCF risk, suggesting putative gene–gene interactions. Among them, the combination of *CD320* C*-TCN2* T*-SLC19A1* G*-SLC19A2* T showed the strongest association with osteoporosis risk. Our analyses therefore suggest that the allelic combinations of *CD320, TCN2, SLC19A1,* and *SLC19A2* genes could play a role in the pathogenesis of osteoporosis and OVCF. 

Currently, there are no genetic studies in 3’–UTR polymorphisms associating osteoporosis with folate and vitamin B12, and only a few clinical studies have evaluated the effects of folic acid and vitamin B12 on plasma homocysteine levels and bone health. In the Hordaland Homocysteine Study, folate was linked to BMD and a reduced fracture risk, but there was limited evidence to support a direct effect of folate on bone health [33,34]. In the current study, we identified significant differences in folate and vitamin B12 levels between the control and osteoporosis groups, although the levels of folate and vitamin B12 were within the normal range for all groups. Even in the stratified analysis, the bottom 15% of folate and vitamin B12 levels were within the normal range. Therefore, we did not draw substantial findings from these genetic analyses. 

We used the LDlink (https://ldlink.nci.nih.gov/) to find out the SNPs linked to each SNPs and identify other SNPs in the linkage disequilibrium (LD) relationship to the SNPS analyzed in the present study. According to the LD link tool (LD proxy), there was no tightly linked variants associated with TCN2 rs10418 C> T and SLC19A2 rs16862199 C> T. By contrast, the CD320 rs9426 C> T and SLC19A1 rs1051296G> T were reported to be tightly linked with CD320 rs2336573 variant and SLC19A2 rs12659, respectively [35,36]. 

There are several limitations of this study. First, the study included only Korean postmenopausal women, and our results cannot be generalized to other racial or ethnic groups because SNPs and allele combinations vary among ethnic groups. Second, this was a hospital–based case–control study, and the sample size was relatively small. Thus, the present results should be replicated and validated in larger studies that include diverse ethnic groups and men. Third, we could not conclusively exclude other potential confounding variables, such as exposure to different environmental factors (e.g., smoking, nutrition, calcium and vitamin intake), and this study was not a genome–wide association study; therefore, the additional genetic risk factors affecting osteoporosis and OVCF are unclear. Fourth, we did not identify the miRNAs of the target genes and did not perform in vitro studies. Lastly, we should consider the possible changes of analyzed genes due to change in miRNA action because we investigated SNPs located in 3´–UTR of B vitamin–related genes. Genetic variation in the 3´–UTR can affect gene expression by interfering with miRNA binding [12,37,38]. Polymorphisms residing within the miRNA–binding site of the target genes may have their own disease onset effect, but changes in the miRNA binding efficiency also has to be considered [23,24]. Thus, our study does not rule out the possibility that change in miRNA action could influence our results.

## 5. Conclusions

We have identified associations between four B vitamin–related SNPs in 3’–UTR, namely *CD320*C>T (rs9426), *TCN2*C>T (rs10418), *SLC19A1*G>T (rs1051296), and *SLC19A2*C>T (rs16862199), and the occurrence of osteoporosis and OVCF in Korean postmenopausal women. Our findings suggest that SNPs in the miRNA binding site within the 3’–UTR of the *CD320* gene may contribute to osteoporosis and OVCF occurrences in some individuals with high homocysteine level. Although these findings do not broadly address the complex pathogenesis of osteoporosis, the data described here could contribute to the available pool of SNP variants needed for the individual assessment of osteoporosis and OVCF risk.

## Figures and Tables

**Figure 1 genes-11-00612-f001:**
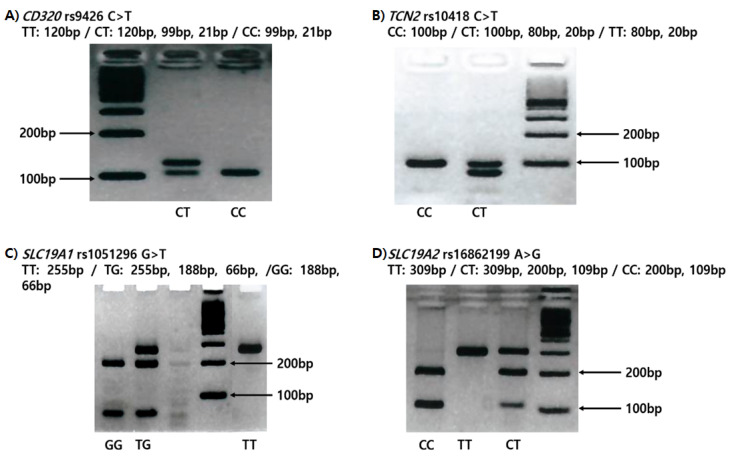
Restriction fragment length polymorphism (RFLP) variations according to genotypes. (**A**) *CD320 rs9426 C>T* (**B**) *TCN2 rs10418 C>T* (**C**) *SLC19A1 rs1051296 G>T* (**D**) *SLC19A2 rs16862199 C>T.*

**Figure 2 genes-11-00612-f002:**
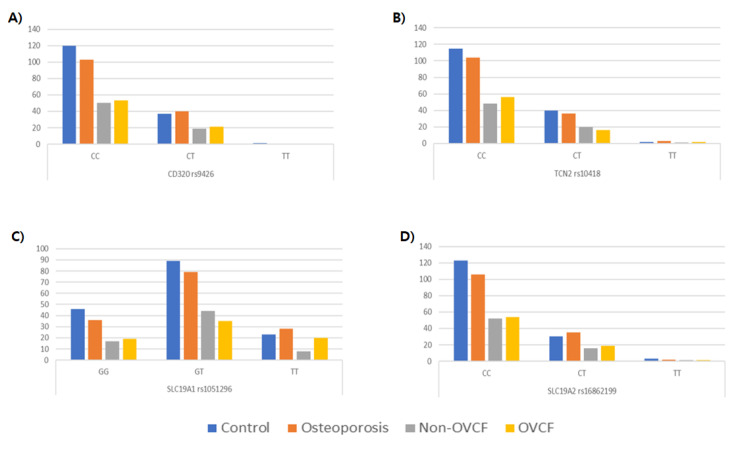
Genotype frequencies of single nucleotide polymorphisms. (**A**) *CD320 rs9426 C>T* (**B**) *TCN2 rs10418 C>T* (**C**) *SLC19A1 rs1051296 G>T* (**D**) *SLC19A2 rs16862199 C>T.*

**Figure 3 genes-11-00612-f003:**
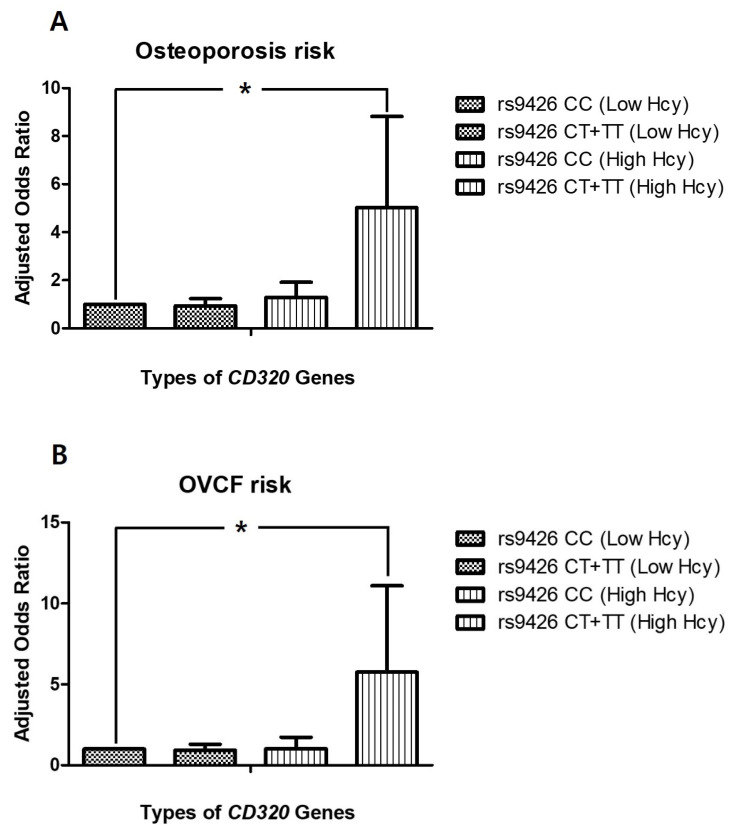
(**A**) Osteoporosis risk stratified by interaction between *CD320 rs9426* C>T and homocysteine levels (**B**) OVCF risk stratified by interaction between *CD320 rs9426* C>T and homocysteine levels. Hcy, homocysteine; OVCF, osteoporotic vertebral compression fracture; * *p* < 0.05.

**Table 1 genes-11-00612-t001:** Baseline characteristics of control and osteoporosis groups.

Characteristic	Control (n = 158)	Osteoporosis (n = 143)	*p **	Non-OVCF (n = 69)	*p ***	OVCF (n = 74)	*p ****
Age (years, mean ± SD)	69.36 ± 6.26	69.38 ± 7.25	0.983	70.83 ± 6.53	0.207	70.23 ± 9.44	0.282
Hypertension (*n*)	79 (50.0%)	52 (36.4%)	0.134	20 (29.0%)	<0.0001	32 (43.2%)	0.0007
SBP (mmHg, mean ± SD)	135.92 ± 18.65	127.32 ± 14.44	<0.0001	126.54 ± 13.76	0.0002	128.13 ± 15.16	0.003
DBP (mmHg, mean ± SD)	80.73 ± 11.52	75.48 ± 10.33	0.0001	74.42 ± 10.24	0.0001	76.57 ± 10.38	0.012
Diabetes mellitus (n)	22 (13.9%)	26 (18.2%)	0.392	17 (24.6%)	0.645	9 (12.2%)	0.05
FBS (mg/dL, mean ± SD)	111.78 ± 28.12	123.56 ± 43.20	0.006	114.70 ± 26.41	0.466	132.56 ± 54.01	0.0002
Homocysteine (μmol/L, mean ± SD)	9.74 ± 3.04	9.83 ± 4.09	0.83	9.54 ± 4.35	0.69	10.11 ± 3.84	0.441
Folate (ng/mL, mean ± SD)	9.51 ± 6.48	8.28 ± 4.87	0.074	10.27 ± 4.95	0.408	6.57 ± 4.11	0.0005
BMI (kg/m^2^, mean ± SD)	24.52 ± 3.11	23.48 ± 3.81	0.045	23.88 ± 2.77	0.173	21.37 ± 6.98	0.005
HDL (mg/dL, mean ± SD)	47.67 ± 12.13	44.78 ± 14.24	0.223	44.42 ± 12.27	0.18	45.16 ± 16.19	0.385
LDL (mg/dL, mean ± SD)	130.30 ± 44.64	107.31 ± 39.55	0.002	95.18 ± 32.38	<0.0001	120.64 ± 42.64	0.281
TG (mg/dL, mean ± SD)	152.85 ± 87.54	146.81 ± 81.90	0.567	133.89 ± 87.61	0.169	159.05 ± 74.81	0.633
Vitamin B12 (pg/mL, mean ± SD)	824.92 ± 919.42	593.13 ± 353.52	0.033	552.61 ± 292.27	0.023	750.13 ± 510.97	0.75
BMD (g/cm^2^, mean ± SD)	-	−3.04 ± 0.94	-	−3.13 ± 0.62	-	−2.93 ± 1.21	-

SD, standard deviation; OVCF, osteoporotic vertebral compression fracture; SBP, systolic blood pressure; DBP, diastolic blood pressure; FBS, fasting blood sugar; BMI, body mass index; HDL, high density lipoprotein; LDL, low density lipoprotein; TG, triglyceride; BMD, bone mineral density. *p* *—Osteoporosis versus Controls, *p* **—Non OVCF vs Controls, and *p* ***—OVCF vs Controls.

**Table 2 genes-11-00612-t002:** Genotype frequencies for the 3’-UTR polymorphisms of *CD320, TCN2, SLC19A1* and *SLC19A2* genes in control and OVCF groups.

Genotype ^†^	Control (n = 158), n (%)	Osteoporosis (n = 143), n (%)	AOR^a^ (95% CI)	*p **	Non–OVCF (n = 69)	AOR (95% CI)	*p ***	OVCF (n = 74)	AOR (95% CI)	*p ****
*CD320*C>T (rs9426)										
CC	120 (75.9)	103 (72.0)	1.000 (reference)		50 (72.5)	1.000 (reference)		53 (71.6)	1.000 (reference)	
CT	37 (23.4)	40 (28.0)	1.256 (0.738–2.138)	0.401	19 (27.5)	1.246 (0.638–2.434)	0.520	21 (28.4)	1.408 (0.731–2.712)	0.307
TT	1 (0.6)	0 (0.0)	N/A	0.998	0 (0.0)	N/A	0.998	0 (0.0)	N/A	0.998
Dominant (CC vs CT+TT)			1.222 (0.720–2.074)	0.459		1.210 (0.621–2.359)	0.576		1.369 (0.712–2.630)	0.347
Recessive (CC+CT vs TT)			N/A	0.998		N/A	0.998		N/A	0.998
HWE–*P*	0.264	0.052								
*TCN2*C>T (rs10418)										
CC	115 (73.2)	104 (72.7)	1.000 (reference)		48 (69.6)	1.000 (reference)		56 (75.7)	1.000 (reference)	
CT	40 (25.5)	36 (25.2)	1.015 (0.593–1.736)	0.957	20 (29.0)	1.342 (0.690–2.610)	0.387	16 (21.6)	0.750 (0.372–1.513)	0.422
TT	2 (1.3)	3 (2.1)	0.853 (0.182–3.993)	0.840	1 (1.4)	0.665 (0.067–6.645)	0.729	2 (2.7)	1.215 (0.211–7.000)	0.828
Dominant (CC vs CT+TT)			1.007 (0.599–1.695)	0.978		1.279 (0.668–2.447)	0.458		0.800 (0.410–1.561)	0.514
Recessive (CC+CT vs TT)			0.884 (0.192–4.077)	0.874		0.611 (0.062–5.998)	0.673		1.372 (0.240–7.841)	0.722
HWE–*P*	0.473	0.955								
*SLC19A1*G>T (rs1051296)										
GG	46 (29.1)	36 (25.2)	1.000 (reference)		17 (24.6)	1.000 (reference)		19 (25.7)	1.000 (reference)	
GT	89 (56.3)	79 (55.2)	1.059 (0.611–1.835)	0.839	44 (63.8)	1.241 (0.623–2.474)	0.539	35 (47.3)	0.868 (0.436–1.727)	0.686
TT	23 (14.6)	28 (19.6)	1.388 (0.653–2.949)	0.394	8 (11.6)	0.853 (0.303–2.401)	0.763	20 (27.0)	1.659 (0.686–4.009)	0.261
Dominant (GG vs GT+TT)			1.098 (0.649–1.858)	0.727		1.138 (0.582–2.227)	0.705		0.990 (0.516–1.898)	0.975
Recessive (GG+GT vs TT)			1.253 (0.658–2.387)	0.493		0.713 (0.290–1.751)	0.460		1.670 (0.796–3.505)	0.175
HWE–*P*	0.058	0.195								
*SLC19A2*C>T (rs16862199)										
CC	123 (78.8)	106 (74.1)	1.000 (reference)		52 (75.4)	1.000 (reference)		54 (73.0)	1.000 (reference)	
CT	30 (19.2)	35 (24.5)	1.395 (0.794–2.451)	0.248	16 (23.2)	1.124 (0.545–2.317)	0.752	19 (25.7)	1.678 (0.842–3.344)	0.142
TT	3 (1.9)	2 (1.4)	0.823 (0.133–5.093)	0.834	1 (1.4)	0.930 (0.090–9.660)	0.952	1 (1.4)	0.831 (0.082–8.417)	0.875
Dominant (CC vs CT+TT)			1.345 (0.778–2.326)	0.288		1.118 (0.554–2.255)	0.756		1.607 (0.822–3.142)	0.166
Recessive (CC+CT vs TT)			0.758 (0.124–4.646)	0.764		0.915 (0.089–9.423)	0.941		0.782 (0.079–7.737)	0.833
HWE–*P*	0.602	0.640								

† Odds ratio adjusted by age, gender, hypertension, diabetes mellitus, BMI, glucose and folate *p* *—Osteoporosis versus Controls, *p* **—Non OVCF vs Controls, and *p* ***—OVCF vs Controls. AOR, adjusted odds ratio; CI, confidence interval; OVCF, osteoporotic vertebral compression fracture, HWE-*P*, Hardy-Weinberg equilibrium p-value; N/A, not applicable.

**Table 3 genes-11-00612-t003:** Allele combination analysis for four vitamin B-related genes (*CD320, TCN2, SLC19A1*, and *SLC19A2*) using the multifactor dimensionality reduction method to compare genotype frequencies between OVCF and control groups.

Allele Combination	Control^†^ (2 n = 316)	Osteoporosis^†^ (2 n = 286)	OR (95% CI)	*p **	Non–OVCF^†^ (2 n = 148)	OR (95% CI)	*p ***	OVCF^†^ (2 n = 138)	OR (95% CI)	*p ****
***CD320* rs9246C>T */ TCN2* rs10418C>T */ RFC* rs1051296G>T */ SLC19A2* rs16862199C>T**										
C-C-G-C	0.390	0.317	1.000 (reference)		0.386	1.000 (reference)		0.269	1.000 (reference)	
C-C-G-T	0.039	0.062	2.027 (0.930–4.420)	0.080	0.039	0.324 (0.324–2.882)	1.000	0.079	1.280 (1.280–7.386)	0.014
C-C-T-C	0.282	0.308	1.336 (0.896–1.995)	0.185	0.266	0.585 (0.585–1.592)	0.899	0.329	1.028 (1.028–2.788)	0.043
C-C-T-T	0.016	0.038	2.974 (0.998–8.858)	0.065	0.024	0.321 (0.321–6.041)	0.701	0.055	1.522 (1.522–15.900)	0.008
C-T-G-C	0.087	0.032	0.451 (0.202–1.005)	0.065	0.023	0.075 (0.075–0.887)	0.025	0.042	0.263 (0.263–1.774)	0.506
C-T-G-T	0.000	0.017	14.850 (0.810–272.100)	0.015	0.028	1.098 (1.098–393.000)	0.009	0.001	N/A	
C-T-T-C	0.033	0.083	3.244 (1.478–7.120)	0.003	0.088	1.133 (1.133–6.844)	0.030	0.083	1.482 (1.482–9.187)	0.005
C-T-T-T	0.031	0.002	0.135 (0.017–1.075)	0.030	0.009	0.029 (0.029–1.860)	0.182	0.000	0.008 (0.008–2.535)	0.119
T-C-G-C	0.042	0.078	2.287 (1.094–4.782)	0.029	0.048	0.472 (0.472–3.309)	0.619	0.102	1.556 (1.556–8.089)	0.003
T-C-G-T	0.006	0.013	2.703 (0.484–15.090)	0.407	0.030	0.824 (0.824–26.130)	0.078	0.000	0.029 (0.029–12.980)	1.000
T-C-T-C	0.040	0.036	1.040 (0.437–2.477)	1.000	0.048	0.472 (0.472–3.309)	0.619	0.030	0.292 (0.292–3.068)	1.000
T-C-T-T	0.026	0.000	0.079 (0.005–1.394)	0.022	0.000	0.008 (0.008–2.397)	0.107	0.000	0.010 (0.010–3.179)	0.200
T-T-G-C	0.006	0.009	2.027 (0.332–12.390)	0.654	0.011	0.318 (0.318–16.920)	0.587	0.000	0.029 (0.029–12.980)	1.000
T-T-T-C	0.002	0.000	0.450 (0.018–11.180)	1.000	0.001	N/A		0.003	0.041 (0.041–25.470)	1.000
T-T-T-T	0.000	0.004	4.049 (0.163–100.600)	0.428	0.000	0.031 (0.031–19.210)	1.000	0.007	0.365 (0.365–229.200)	0.250

† The allele combination models were indicated by frequency. *p*-value calculated by Fisher’s exact test. *p* *—Osteoporosis versus Controls, *p* **—Non OVCF vs Controls, and *p* ***—OVCF vs Controls. OR, odds ratio; CI, confidence interval; OVCF, osteoporotic vertebral compression fracture.

**Table 4 genes-11-00612-t004:** Stratified analysis of osteoporosis incidence by interactions with age, hypertension, diabetes mellitus, and levels of vitamin B12, folate, and homocysteine.

Variables^†^	*CD320 rs9426* CC vs. CT+TT		*TCN2 rs10418* CC vs. CT+TT		*SLC19A1 rs1051296* GG vs. GT+TT		*SLC19A2 rs16862199* CC vs. CT+TT	
	AOR (95% CI)	AOR (95% CI)	AOR (95% CI)	AOR (95% CI)	AOR (95% CI)	AOR (95% CI)	AOR (95% CI)	AOR (95% CI)
Age (years)								
<69	1.000 (reference)	0.698 (0.316–1.540)	1.000 (reference)	0.944 (0.486–1.835)	1.000 (reference)	2.731 (1.170–6.375)	1.000 (reference)	1.354 (0.577–3.176)
≥69	1.050 (0.607–1.818)	2.399 (1.101–5.226)	0.936 (0.479–1.828)	1.848 (0.985–3.469)	4.105 (1.574–10.708)	2.740 (1.179–6.365)	1.361 (0.791–2.344)	1.943 (0.917–4.119)
Hypertension								
No	1.000 (reference)	1.685 (0.832–3.412)	1.000 (reference)	2.210 (1.193–4.092)	1.000 (reference)	1.372 (0.674–2.792)	1.000 (reference)	2.488 (1.119–5.531)
Yes	0.766 (0.441–1.328)	0.631 (0.272–1.465)	1.000 (0.510–1.960)	0.775 (0.397–1.513)	0.783 (0.325–1.884)	0.703 (0.332–1.491)	0.839 (0.486–1.450)	0.561 (0.245–1.288)
Diabetes mellitus								
No	1.000 (reference)	1.330 (0.744–2.376)	1.000 (reference)	1.191 (0.724–1.959)	1.000 (reference)	1.293 (0.731–2.286)	1.000 (reference)	1.655 (0.891–3.076)
Yes	1.726 (0.828–3.598)	1.721 (0.553–5.362)	0.800 (0.334–1.914)	3.413 (1.245–9.358)	1.810 (0.600–5.458)	1.595 (0.681–3.738)	2.111 (0.978–4.555)	1.193 (0.422–3.370)
Vitamin B12^a^								
>395 pg/mL	1.000 (reference)	1.273 (0.738–2.196)	1.000 (reference)	1.200 (0.678–2.125)	1.000 (reference)	1.256 (0.723–2.182)	1.000 (reference)	1.200 (0.678–2.125)
≤395 pg/mL	2.765 (1.198–6.382)	3.437 (0.646–18.278)	2.164 (0.936–5.004)	4.454 (0.893–22.208)	3.099 (0.713–13.470)	2.442 (0.939–6.353)	2.164 (0.936–5.004)	4.454 (0.893–22.208)
Folate^a^								
>4.59 nmol/L	1.000 (reference)	1.325 (0.742–2.364)	1.000 (reference)	1.352 (0.826–2.212)	1.000 (reference)	1.000 (0.248–4.033)	1.000 (reference)	1.467 (0.788–2.734)
≤4.59 nmol/L	1.961 (0.907–4.238)	2.157 (0.576–8.082)	1.392 (0.517–3.752)	2.475 (1.014–6.041)	1.000 (0.248–4.033)	2.019 (0.878–4.646)	2.032 (0.906–4.561)	1.837 (0.582–5.796)
Homocysteine^a^								
<12.68 μmol/L	1.000 (reference)	0.936 (0.513–1.710)	1.000 (reference)	1.646 (1.001–2.707)	1.000 (reference)	1.484 (0.842–2.615)	1.000 (reference)	1.591 (0.857–2.954)
≥12.68 μmol/L	1.293 (0.547–3.055)	5.019 (1.533–16.430)	3.411 (1.213–9.595)	1.837 (0.783–4.312)	5.385 (1.355–21.400)	1.831 (0.783–4.280)	2.646 (1.141–6.132)	1.569 (0.509–4.838)

^†^ Odds ratio adjusted by age, gender, hypertension, and diabetes mellitus. ^a^ Cut–offs were set at the bottom 15% for vitamin B12 (395 pg/mL) and folate (4.59 nmol/L) and at the top 15% for homocysteine (12.68 umol/L) for both osteoporosis patients and controls.

**Table 5 genes-11-00612-t005:** Stratified analysis of OVCF incidence by interactions with age, hypertension, diabetes mellitus, and levels of vitamin B12, folate, and homocysteine.

Variables^†^	*CD320 rs9426* CC vs. CT+TT		*TCN2 rs10418* CC vs. CT+TT		*SLC19A1 rs1051296* GG vs. GT+TT		*SLC19A2 rs16862199* CC vs. CT+TT	
	AOR (95% CI)	AOR (95% CI)	AOR (95% CI)	AOR (95% CI)	AOR (95% CI)	AOR (95% CI)	AOR (95% CI)	AOR (95% CI)
Age								
<69	1.000 (reference)	0.383 (0.133––1.104)	1.000 (reference)	1.257 (0.479–3.301)	1.000 (reference)	3.782 (1.210–11.817)	1.000 (reference)	1.257 (0.479–3.301)
≥69	0.642 (0.334–1.234)	2.101 (0.907–4.868)	1.057 (0.557–2.004)	1.489 (0.638–3.473)	5.132 (1.468–17.939)	2.578 (0.810–8.208)	1.057 (0.557–2.004)	1.489 (0.638–3.473)
Hypertension								
No	1.000 (reference)	1.485 (0.640–3.446)	1.000 (reference)	2.125 (0.829–5.448)	1.000 (reference)	1.268 (0.517–3.110)	1.000 (reference)	2.125 (0.829–5.448)
Yes	0.911 (0.469–1.768)	0.966 (0.372–2.512)	0.900 (0.469–1.726)	0.963 (0.392–2.367)	0.879 (0.289–2.670)	0.938 (0.372–2.367)	0.900 (0.469–1.726)	0.963 (0.392–2.367)
Diabetes mellitus								
No	1.000 (reference)	1.270 (0.646–2.496)	1.000 (reference)	1.781 (0.881–3.600)	1.000 (reference)	1.257 (0.630–2.510)	1.000 (reference)	1.781 (0.881–3.600)
Yes	0.851 (0.310–2.340)	1.273 (0.301–5.379)	1.301 (0.486–3.477)	0.543 (0.113–2.621)	1.492 (0.368–6.038)	0.847 (0.261–2.744)	1.301 (0.486–3.477)	0.543 (0.113–2.621)
Vitamin B12^a^								
>395 pg/mL	1.000 (reference)	1.242 (0.653–2.362)	1.000 (reference)	1.259 (0.646–2.454)	1.000 (reference)	1.064 (0.561–2.018)	1.000 (reference)	1.259 (0.646–2.454)
≤395 pg/mL	0.845 (0.215–3.313)	1.524 (0.129–18.010)	0.469 (0.098–2.255)	2.495 (0.331–18.782)	N/A	0.784 (0.201–3.056)	0.469 (0.098–2.255)	2.495 (0.331–18.782)
Folate^a^								
>4.59 nmol/L	1.000 (reference)	1.097 (0.520–2.315)	1.000 (reference)	1.575 (0.740–3.351)	1.000 (reference)	1.028 (0.498–2.120)	1.000 (reference)	1.575 (0.740–3.351)
≤4.59 nmol/L	3.219 (1.418–7.307)	7.307 (1.975–27.033)	4.290 (1.911–9.629)	3.085 (0.914–10.411)	2.022 (0.469–8.730)	3.589 (1.440–8.950)	4.290 (1.911–9.629)	3.085 (0.914–10.411)
Homocysteine^a^								
<12.68 μmol/L	1.000 (reference)	0.931 (0.456–1.900)	1.000 (reference)	1.810 (0.898–3.649)	1.000 (reference)	1.129 (0.570–2.237)	1.000 (reference)	1.810 (0.898–3.649)
≥12.68μmol/L	1.020 (0.335–3.103)	5.760 (1.480–22.417)	2.911 (1.100–7.704)	0.861 (0.168–4.425)	3.100 (0.492–19.542)	1.286 (0.434–3.807)	2.911 (1.100–7.704)	0.861 (0.168–4.425)

† Odds ratio adjusted by age, gender, hypertension, and diabetes mellitus. OVCF, osteoporotic vertebral compression fracture. ^a^ Cut–offs were set at the bottom 15% for vitamin B12 (395 pg/mL) and folate (4.59 nmol/L) and at the top 15% for homocysteine (12.68 umol/L) for both osteoporosis patients and controls.

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
