# Peer review of "3’-UTR Polymorphisms of Vitamin B-Related Genes Are Associated with Osteoporosis and Osteoporotic Vertebral Compression Fractures (OVCFs) in Postmenopausal Women"

_genes, 2020, doi:10.3390/genes11060612_

Round 1
Reviewer 1 Report
Tae-Keun Ahn and colleagues presented here article entitled "3'-UTR polymorphisms of vitamin B-related genes (TCblRC>T, 2 TCN2C>T, RFC1G>T, and TC1C>T) associated with osteoporosis and osteoporotic vertebral compression fractures (OVCFs) in postmenopausal women" that raises the issue of susceptibility to menopausal osteoporosis and bone fractures which is a major medical and social problem.
The authors performed an analysis of 4 SNPs located in 3'-UTR of genes involved in the metabolism of vitamins B. They chose analyzed variants based on bioinformatic analysis with MicroSNiPer to identify potential miRNA binding sites within the 3'-UTRs.
In my opinion, this is an innovatory approach that makes the article original.
Nonetheless, I have several issues that must be regarded.
For first, the title should be corrected. It should be: 3'-UTR polymorphisms of vitamin B-related genes are (TCblRC>T, 2 TCN2C>T, RFC1G>T, and TC1C>T) associated with osteoporosis and osteoporotic vertebral compression fractures (OVCFs) in postmenopausal women.
The terminology of SNPs listed in the title is incorrect.
For example, it should be listed as CD320 c.189C>T or using rs number.
Moreover, the names of the genes should also be corrected according to the HGNC database:
TCblR as CD320,
RFC1 as SLC19A1
TC1 as SLC19A2
The terminology should be adjusted in the whole article.
In Abstract authors should provide P-values for AORs in lines 28 and 31.
The Introduction section should be more extensive. This part should contain a clear justification for the research undertaken. The authors included this justification in principle in one short paragraph, which, in my opinion, does not adequately explain why the study described was undertaken.
The authors should bear in mind that the reader does not know the details of the metabolism of B vitamins and should be interested in why polymorphisms in regulatory sequences of vitamin B-related genes can significantly change the expression of these genes and thus bone health.
Please write it more clearly and comprehensively.
Materials and methods are well described. The only remark I have is the lack of a DNA extraction method in section 2.4. "Genetic analyses".
In the Results section, in point 3.1. authors described patients' group. However, it was reported somewhere else, and it should be mentioned by proper reference [Ahn TK, Kim JO, Kim HW, et al. 3'-UTR Polymorphisms of MTHFR and TS Associated with Osteoporotic Vertebral Compression Fracture Susceptibility in Postmenopausal Women. Int J Mol Sci. 2018;19(3):824. DOI:10.3390/ijms19030824].
In points 3.3. "Allele combination analysis using the MDR method" and 3.4. "Stratified and interactions analyses between TCblR, TCN2, RFC1, and TC1 SNPs and clinical parameters" authors listed ORs with confidence intervals but not P-values – they should be listed too.
I also have small critical remarks to Tables that are readable but may be improved in my opinion.
Tables 1-3 are constructed similarly and in columns 4, 6 and 8 are listed P-values. Nonetheless, I would only mark the P values in the following way: P* or P1, P** or P2, and P*** or P3, adding the description below the table: P* - Osteoporosis versus Controls, P** - Non OVCF vs Controls, and P*** - OVCF vs Controls.
Table 4 and 5 missing P-values – why?
The Discussion section is, in my opinion, a little too vague. The few publications on the subject may justify it, but authors might regard here the possible effect of expression changes of analyzed genes, due to possible miRNA action.
I have one more suggestion: to include a short sentence in the discussion section, including information that the studied SNPs may also be in linkage disequilibrium with different unidentified sequence variants and thus be associated with osteoporosis and risk fractures. Such information would be useful to a reader who is a clinician and not a geneticist.
In lines: 270-271 authors write: "These results highlight the influence of genotype on the incidence of osteoporosis and OVCF in individuals with high homocysteine levels;…"
I would change the word "influence" for "relationship", which is more liberal.
In line 277 there is: "in a subset of patients that are genotype…" it would be better "in a subset of patients, who carry/carrying genotype…"
Overall the presented study is interesting.
Author Response
We would like to thank you and the reviewers for considering our manuscript “3’-UTR polymorphisms of vitamin B-related genes (TCblRC>T, TCN2C>T,
RFC1G>T, and TC1C>T) associated with osteoporosis and osteoporotic vertebral
compression fractures (OVCFs) in postmenopausal women " (genes-777479) and providing us the valuable feedback. We have revised our manuscript according to the reviewer’s comments as detailed below. Moreover, a list of changes is highlighted in red in the revised manuscript.
Comment
For first, the title should be corrected. It should be: 3'-UTR polymorphisms of vitamin B-related genes are (TCblRC>T, 2 TCN2C>T, RFC1G>T, and TC1C>T) associated with osteoporosis and osteoporotic vertebral compression fractures (OVCFs) in postmenopausal women.
Response:
As suggested by the reviewer, we have changed the title.
Comment
The terminology of SNPs listed in the title is incorrect.
For example, it should be listed as CD320 c.189C>T or using rs number. Moreover, the names of the genes should also be corrected according to the HGNC database: The terminology should be adjusted in the whole article.
Response:
As suggested by the reviewer, we have changed the terminology.
Comment
In Abstract authors should provide P-values for AORs in lines 28 and 31.
Response:
As suggested by the reviewer, we have added P-values.
Comment
The Introduction section should be more extensive. This part should contain a clear justification for the research undertaken. The authors included this justification in principle in one short paragraph, which, in my opinion, does not adequately explain why the study described was undertaken.
The authors should bear in mind that the reader does not know the details of the metabolism of B vitamins and should be interested in why polymorphisms in regulatory sequences of vitamin B-related genes can significantly change the expression of these genes and thus bone health.
Response:
As suggested by the reviewer, we have added a clear justification for the research undertaken. Also we have added B vitamin metabolisms and possible effect of SNPs on bone health in Introduction and Discussion.
Comment
Materials and methods are well described. The only remark I have is the lack of a DNA extraction method in section 2.4. "Genetic analyses".
Response:
As suggested by the reviewer, we have added Genetic analyses in Methods.
Comment
In the Results section, in point 3.1. authors described patients' group. However, it was reported somewhere else, and it should be mentioned by proper reference [Ahn TK, Kim JO, Kim HW, et al. 3'-UTR Polymorphisms of MTHFR and TS Associated with Osteoporotic Vertebral Compression Fracture Susceptibility in Postmenopausal Women. Int J Mol Sci. 2018;19(3):824. DOI:10.3390/ijms19030824].
Response:
As suggested by the reviewer, we have added reference for that.
Comment
In points 3.3. "Allele combination analysis using the MDR method" and 3.4. "Stratified and interactions analyses between TCblR, TCN2, RFC1, and TC1 SNPs and clinical parameters" authors listed ORs with confidence intervals but not P-values – they should be listed too.
Response:
As suggested by the reviewer, we have added P-values.
Comment
I also have small critical remarks to Tables that are readable but may be improved in my opinion.
Tables 1-3 are constructed similarly and in columns 4, 6 and 8 are listed P-values. Nonetheless, I would only mark the P values in the following way: P* or P1, P** or P2, and P*** or P3, adding the description below the table: P* - Osteoporosis versus Controls, P** - Non OVCF vs Controls, and P*** - OVCF vs Controls.
Response:
As suggested by the reviewer, we have added footnote described explain P-value follow as; P* - Osteoporosis versus Controls, P** - Non OVCF vs Controls, and P*** - OVCF vs Controls.
Comment
Table 4 and 5 missing P-values – why?
Response:
We appreciate your valuable comment. Table 4 and Table 5 are to investigate the synergy effect of the 'Odd ratio' in the group that is set as a risk group for each condition. The significance of the statistical value can be confirmed by the '95% CI' value. When the P-value is presented, the purpose of the explanation in the table may be blurred because it is focused on how much the p-value is lower than the synergy of odd ratio.
Comment
The Discussion section is, in my opinion, a little too vague. The few publications on the subject may justify it, but authors might regard here the possible effect of expression changes of analyzed genes, due to possible miRNA action.
Response:
We have added following paragraph in Discussion.
Lastly, we should consider the possible changes of analyzed genes due to change in miRNA action because we investigated SNPs located in 3´-UTR of B vitamin-related genes. Genetic variation in the 3´-UTR can affect gene expression by interfering with miRNA binding [Ref1-Ref 3]. Polymorphisms residing within the miRNA-binding site of the target genes may have their own disease onset effect, but changes in the miRNA binding efficiency also has to be considered [11,21,22]. Thus, our study does not rule out the possibility that change in miRNA action could influence our results.
Ref 1.
3'-UTR Polymorphisms in the Vascular Endothelial Growth Factor Gene (VEGF) Contribute to Susceptibility to Recurrent Pregnancy Loss (RPL). An HJ, Kim JH, Ahn EH, Kim YR, Kim JO, Park HS, Ryu CS, Kim EG, Cho SH, Lee WS, Kim NK. Int J Mol Sci. 2019 Jul 5;20(13). pii: E3319. doi: 10.3390/ijms20133319. PMID: 31284523
Ref 2.
3'-UTR Polymorphisms of MTHFR and TS Associated with Osteoporotic Vertebral Compression Fracture Susceptibility in Postmenopausal Women. Ahn TK, Kim JO, Kim HW, Park HS, Shim JH, Ropper AE, Han IB, Kim NK. Int J Mol Sci. 2018 Mar 12;19(3). pii: E824. doi: 10.3390/ijms19030824. PMID: 29534533
Ref 3.
Interplay between 3'-UTR polymorphisms in the methylenetetrahydrofolate reductase (MTHFR) gene and the risk of ischemic stroke. Kim JO, Park HS, Ryu CS, Shin JW, Kim J, Oh SH, Kim OJ, Kim NK. Sci Rep. 2017 Sep 29;7(1):12464. doi: 10.1038/s41598-017-12668-x. PMID: 28963520
Comment
I have one more suggestion: to include a short sentence in the discussion section, including information that the studied SNPs may also be in linkage disequilibrium with different unidentified sequence variants and thus be associated with osteoporosis and risk fractures. Such information would be useful to a reader who is a clinician and not a geneticist
Response:
We used the LD link (https://ldlink.nci.nih.gov/) to find out the SNPs linked to each SNPs to identify other SNPs in the LD relationship to the SNPs analyzed in the study.
Comment
In lines: 270-271 authors write: "These results highlight the influence of genotype on the incidence of osteoporosis and OVCF in individuals with high homocysteine levels;…"
I would change the word "influence" for "relationship", which is more liberal.
Response:
As suggested by the reviewer, we corrected it.
Comment
In line 277 there is: "in a subset of patients that are genotype…" it would be better "in a subset of patients, who carry/carrying genotype…"
Response:
As suggested by the reviewer, we corrected it.

Reviewer 2 Report
This clinical research tries to correlate 4 SNPs to osteoporosis and OVCF. It is always important to find new SNPs correlated to osteoporosis and OVCF, which is common in Asian women. The rationale to choose those 4 genes and 4 SNPs needs to be clarified in the introduction.
Concerns:
Four SNPs are chosen in this research without rationale. The rationale to choose these 4 genes and 4 SNPs, the methods to find these 4 SNPs may help to understand the purpose of the research. Please explain why these 4 genes are so important in the pathway and why these 4 SNPs is important for these 4 genes.
DNA seq is very cheap and quick now, so it's highly questionable why RFLP is still performed in this research instead of direct sequencing. It is important to explain why RFLP is better than sequencing.
There is no explanation about those "TCblRC>T", which should be explained before use. "TCblRC>T" should always be written as "TCblR C>T" to seperate the gene name and SNP name. "C", "T", "CC", and "TT"themselves should also be defined based on your RFLP bands before use. These are not straight-forward and make the paper very hard to read.
Please make a figure showing the RFLP results and bar chart about results mentioned in result 3.2 and 3.3.
Please also explain the rationale to check the allele combinations. Is there biological meaning or genetic connections of those combinations?
The results in 3.4 are impressive. Is there correlation between homocysteine level and osteoporotic chance in CT+TT people? Bar chart is necessary for all the results not just those significant results. Star can be used to highlight the significance, and P value should be labeled on the bar chart or mentioned in the figure legend.
Author Response
We would like to thank you and the reviewers for considering our manuscript “3’-UTR polymorphisms of vitamin B-related genes (TCblRC>T, TCN2C>T,
RFC1G>T, and TC1C>T) associated with osteoporosis and osteoporotic vertebral
compression fractures (OVCFs) in postmenopausal women " (genes-777479) and providing us the valuable feedback. We have revised our manuscript according to the reviewer’s comments as detailed below. Moreover, a list of changes is highlighted in red in the revised manuscript.
---------------------------------------------
Reviewer #2:
Comment
This clinical research tries to correlate 4 SNPs to osteoporosis and OVCF. It is always important to find new SNPs correlated to osteoporosis and OVCF, which is common in Asian women. The rationale to choose those 4 genes and 4 SNPs needs to be clarified in the introduction.
Response:
As suggested by the reviewer, we have added the rationale for the research.
Comment
Four SNPs are chosen in this research without rationale. The rationale to choose these 4 genes and 4 SNPs, the methods to find these 4 SNPs may help to understand the purpose of the research. Please explain why these 4 genes are so important in the pathway and why these 4 SNPs is important for these 4 genes
Response:
As suggested by the reviewer, we have added the rationale for choosing four genes.
Comment
DNA seq is very cheap and quick now, so it's highly questionable why RFLP is still performed in this research instead of direct sequencing. It is important to explain why RFLP is better than sequencing
Response:
RFLP is a classic genotyping technique, but its cost is low than DNA seq. and it is easy to carry out large numbers at once. But you need to check its accuracy. In this study, 10% of RFLP results were randomly selected in addition to RFLP alone, and reliability was verified by verifying that the genotype matched 100% by the Sanger sequencing method.
Comment
There is no explanation about those "TCblRC>T", which should be explained before use. "TCblRC>T" should always be written as "TCblR C>T" to seperate the gene name and SNP name. "C", "T", "CC", and "TT"themselves should also be defined based on your RFLP bands before use. These are not straight-forward and make the paper very hard to read.
Response:
As suggested by the reviewer, we have corrected them.
Comment
Please make a figure showing the RFLP results and bar chart about results mentioned in result 3.2 and 3.3.
Response:
As suggested by the reviewer, we have made the figures.
Comment
Please also explain the rationale to check the allele combinations. Is there biological meaning or genetic connections of those combinations?
Response:
There is an analysis that defines the shape of LD and combination groups with the concept of genetically haplotype. However, this is to explain gene mutations that exist on the same chromosome. However, the allele combination is an analysis of allele combinations that may exist, and the number of all cases cannot be considered. It is a method of calculating and suggesting a disease association based on this.
Comment
The results in 3.4 are impressive. Is there correlation between homocysteine level and osteoporotic chance in CT+TT people? Bar chart is necessary for all the results not just those significant results. Star can be used to highlight the significance, and P value should be labeled on the bar chart or mentioned in the figure legend
Response:
The graph of the content is presented in figure 1.
